# Irrigation Strategies with Controlled Water Deficit in Two Production Cycles of Cotton

**DOI:** 10.3390/plants12162892

**Published:** 2023-08-08

**Authors:** Wellinghton Alves Guedes, Reginaldo Gomes Nobre, Lauriane Almeida dos Anjos Soares, Geovani Soares de Lima, Hans Raj Gheyi, Pedro Dantas Fernandes, Ana Paula Nunes Ferreira, André Alisson Rodrigues da Silva, Carlos Alberto Vieira de Azevedo, Daniel Valadão Silva, José Francismar de Medeiros

**Affiliations:** 1Postgraduate Program in Soil and Water Management, Federal Rural University of the Semi-Arid Region, Mossoró 59780-000, RN, Brazil; wellinghton_guedes@hotmail.com (W.A.G.); paulanf18@gmail.com (A.P.N.F.); daniel.valadao@ufersa.edu.br (D.V.S.); jfmedeiros@ufersa.edu.br (J.F.d.M.); 2Academic Unit of Agrarian Sciences, Federal University of Campina Grande, Pombal 58840-000, PB, Brazil; lauriane.almeida@professor.edu.br; 3Academic Unit of Agricultural Engineering, Federal University of Campina Grande, Campina Grande 58430-380, PB, Brazil; geovani.soares@professor.ufcg.edu.br (G.S.d.L.); hans.gheyi@ufcg.edu.br (H.R.G.); pedro.dantas@professor.ufcg.edu.br (P.D.F.); andre.alisson@estudante.ufcg.edu.br (A.A.R.d.S.); carlos.alberto@professor.ufcg.edu.br (C.A.V.d.A.)

**Keywords:** *Gossypium hirsutum* L., water scarcity, phenological stages, water relations

## Abstract

Water scarcity is one of the main abiotic factors that limit agricultural production. In this sense, the identification of genotypes tolerant to water deficit associated with irrigation management strategies is extremely important. In this context, the objective of this study was to evaluate the morphology, production, water consumption, and water use efficiency of colored fiber cotton genotypes submitted to irrigation strategies with a water deficit in the phenological phases. Two experiments were conducted in succession. In the first experiment, a randomized block design was used in a 3 × 7 factorial scheme, corresponding to three colored cotton genotypes (BRS Rubi, BRS Jade, and BRS Safira) in seven irrigation management strategies with 40% of the real evapotranspiration (ETr) varying the phenological stages. In the second experiment, the same design was used in a 3 × 10 factorial arrangement (genotypes × irrigation management strategies). The water deficit in the vegetative phase can be used in the first year of cotton cultivation. Among the genotypes, ‘BRS Jade’ is the most tolerant to water deficit in terms of phytomass accumulation and fiber production.

## 1. Introduction

Water stress is one of the main abiotic stresses limiting agricultural growth and production worldwide [1], particularly in arid and semi-arid regions. The climate changes that occur due to the greenhouse effect have caused a reduction in precipitation and more extreme and severe events, intensifying the effects of water stress [2,3]. Water scarcity has a negative impact on several physiological processes, including germination, photosynthesis, absorption, and/or transport of nutrients and protein structure, which results in a hormonal and nutritional imbalance in plants [4,5,6]. Cotton (*Gossypium hirsutum* L.) is a species belonging to the Malvaceae family, and it is considered the most important source of natural fiber for the textile industry [7,8]. Brazil is among the five largest cotton producers in the world, along with China, India, the United States, and Pakistan, occupying first place in dryland production [9]. In the 2021/2022 season, Brazil produced 21,737 thousand tons of lint cotton, with the states of Mato Grosso and Bahia standing out with cotton yields of 14,506 and 4831 thousand tons, respectively; in this same season, the Northeast region produced 3.20 thousand tons [9,10].

Despite the prominence of cotton production, its cultivation is subjected to various environmental stresses, especially water scarcity, with limitations in agricultural productivity especially in semi-arid regions. Thus, one of the practices that have been widespread to reduce water expenditure in irrigation is the use of controlled water deficit, which consists of reducing the water depth in specific periods during the crop cycle, thereby not compromising the production of the crop [11,12].

In addition, the tolerance of plants to biotic and abiotic stresses depends on the genetic variability existing in the species [13], which is associated with the activation of epigenetic mechanisms during plant development in response to these environmental stimuli, that is, the regulation of stress-responsive genes without changes in the DNA sequence [14,15,16]. Such information represents transcriptional memory and can be triggered in the genome at any stage of plant development [15].

The effect of water deficit on cotton plants can cause changes at all levels of cellular organization. In physiology, for example, it induces the plant to accumulate abscisic acid, which is involved in stomatal closure, reducing photosynthesis and most gas exchange processes [17,18]. As a consequence, there are also reductions in cell division and expansion, formation, and growth of structures such as leaves and stems, as well as biochemical changes in plants, in addition to interference in water potential and induction of accelerated senescence and abscission of leaves [19,20].

Although studies have already been carried out on the tolerance of cotton to water stress, due to its importance for the adoption of adequate irrigation strategies for production under conditions of water scarcity, it is necessary to identify the phenological phases in which the crop is more tolerant or sensitive to water deficit. The reference values of tolerance to water deficit in each development phase and the recovery of the plants in the phenological phases following water stress, as well as the reflexes of the cumulative effect of stress in successive cycles of cultivation, need to be conveniently researched. This study is based on the hypothesis that deficit irrigation management strategies at different phenological stages induce water deficit tolerance in colored fiber cotton genotypes, resulting in increased growth and yield components. In addition, the seeds of these genotypes in a new production cycle will be more tolerant to water deficit. It is worth mentioning that the use of naturally colored fiber cotton genotypes can reduce production costs in the textile industry, since there is no need for dyeing, consequently reducing damage to human health and the environment [21,22].

In this context, the objective of this study was to evaluate the morphology, production, and water relations of genotypes of colored-fiber cotton subjected to irrigation strategies with water deficit in the phenological stages, as well as the influence of water deficit on seeds of these genotypes in a new production cycle, using irrigation management strategies.

## 2. Results and Discussion

There were significant effects of the interaction between the factors of water deficit management strategies and genotypes (*p* ≤ 0.01) on the dry mass of leaf and stem in Experiments I and II and on the number of leaves only in Experiment II (Table 1). There were individual significant effects of water deficit management strategies and cotton genotypes on all variables in both experiments, except for irrigation management strategies for leaf area (Experiment I) and for the number of leaves and leaf area for cotton genotypes in Experiment II (Table 1).

Based on the results of the means comparison test of irrigation management strategies for plant height in Experiment I (Figure 1A), it is observed that the strategies T_1_ (A_1_B_1_C_1_), T_2_ (A_2_B_1_C_1_), T_4_ (A_1_B_1_C_2_), T_6_ (A_2_B_2_C_1_), and T_7_ (A_1_B_2_C_2_) were statistically superior to the others, with PH values of 69.7, 69.5, 73.1, 70.5, and 71.1 cm, respectively, not differing statistically from each other. However, the strategies T_3_ (A_1_B_2_C_1_) and T_5_ (A_2_B_1_C_2_) resulted in the lowest PH values (67.5 and 65.26 cm), with decreases of 3.2 and 6.37% when compared to plants irrigated with 100% ETr (T_1_) throughout the cycle.

In Experiment II, the plant height values obtained in the irrigation strategies T_2_ (B-E0), T_4_ (B-EFL), T_5_ (C-E0), T_7_ (C-EFR), T_8_ (BC-E0), and T_10_ (BC-EFF) did not differ statistically from those of plants irrigated with 100% ETr during the entire cycle, T_1_ (A-E0) (Figure 1C). The water deficit applied in these development stages possibly did not interfere in the growth of cotton since these plants were formed from seeds that had already been exposed to water deficit conditions in Experiment I in these same stages. This may be associated with the predisposition to the expression of genes associated with improvements in tolerance to water stress. Unlike the first experiment, plants subjected to water deficit in the vegetative stage, T_3_ (B-EV), T_6_ (C-EV) and T_9_ (BC-EV), had reductions in plant height of 16.29, 9.84, and 14.82% compared to plants under management with 100% ETr, T_1_ (A-E0), respectively.

The seeds of the second cycle came from seeds produced under water stress in the vegetative, flowering, and flowering/yield formation stages, which led to reductions in plant height. According to Sun et al. [14], the stress memory in plants allows them to respond to stress more efficiently, not only for the plant currently in situations of stress, but also for its future generation. Thus, it can be observed that cotton plants showed improvements in PH during the flowering stage, the yield formation stage, and, successively, in flowering and yield formation stages in the second generation.

For plant height as a function of cotton genotypes in both experiments, the highest values were found in the genotypes ‘BRS Rubi’ and ‘BRS Safira’, with 73.34 and 74.52 cm in the first experiment and 72.98 and 75.28 cm in the second experiment, surpassing ‘BRS Jade’ (Figure 1B,D). These distinctions in plant height can be explained by the genetic variations of the plant materials used, with different genetic constitutions, leading to growth differences between the cotton genotypes [23,24].

In Experiment I, cotton plants subjected to water deficit during the flowering stage showed increments in stem diameter under T_3_ (A_1_B_2_C_1_) and T_7_ (A_1_B_2_C_2_), equal to 3.45 and 4.08%, respectively, when compared to plants under the strategy with full irrigation; another relevant aspect was the recovery in terms of stem diameter in plants that were under water stress in the flowering stage (T_3_—A_1_B_2_C_1_), which was soon after the end of water deficit application (Figure 2A). Cordão et al. [25] observed that water deficit alters the morphology of the plant as it reduces its height, stem diameter, leaf area, and biomass, and it can be said that cotton growth and development are influenced by water deficit in the phenological stages and by cultivars. In this study, it was found that the flowering and yield formation stages for the genotypes are the most sensitive to water deficit. According to Zonta et al. [26], in these stages, cotton plants have greater losses of growth due to the high water requirement during their cycle.

The stem diameter (SD) in Experiment II followed the same trend observed for plant height in relation to water deficit, with higher means of SD obtained with water deficit in the flowering and yield formation stages, according to the strategies T_4_ (B-EFL), T_7_ (C-EFR), and T_10_ (BC-EFF), with SD of 11.22, 10.64, and 11.10 mm, respectively, not differing from the strategies with no water deficit throughout the cycle, namely T_1_ (A-E0), T_2_ (B-E0), T_5_ (C-E0), and T_8_ (BC-E0) (Figure 2C). There was also a reduction in the SD of cotton plants irrigated with 40% ETr in the vegetative stage, T_3_ (B-EV) and T_6_ (C-EV), equal to 10.74 and 10.29%, when compared to plants under full irrigation, respectively, repeating the same trend observed in the first experiment (Figure 2C).

The genotype ‘BRS Jade’ had a larger stem diameter (10.85 mm), regardless of the water deficit management strategy in Experiment I, surpassing ‘BRS Rubi’ and ‘BRS Safira’ by 9.30 and 6.82%, respectively (Figure 2B). In Experiment II, the genotypes ‘BRS Jade’ (11.08 mm) and ‘BRS Safira’ (10.70 mm) did not differ statistically from each other, with increases of 6.4 and 3.08%, respectively, compared to ‘BRS Rubi’ (Figure 2D). Studies on the selection of genotypes of herbaceous cotton have found significant differences between the cultivars regarding the variables analyzed. These variations in means are linked to genetic variation, which exists naturally among genotypes [25].

In Experiment I, the highest NL were observed in the strategies T_1_ (A_1_B_1_C_1_), T_3_ (A_1_B_2_C_1_), T_4_ (A_1_B_1_C_2_), T_6_ (A_2_B_2_C_1_), and T_7_ (A_1_B_2_C_2_), which did not differ statistically from each other, surpassing the strategies T_2_ (A_2_B_1_C_1_) and T_5_ (A_2_B_1_C_2_), which showed reductions of 12.75 and 17.95% when compared to T_1_ (A_1_B_1_C_1_), respectively (Figure 3A). Cotton tolerance to water stress depends on the phenological stage of the plant, and the yield formation stage is more sensitive to water deficit due to impairments of the metabolic and physiological functions of plants directly affecting the number of leaves [27,28].

When comparing the cotton genotypes in Experiment I (Figure 3B), it was observed that there was no statistical difference between ‘BRS Rubi’ (44.03 leaves) and ‘BRS Jade’ (45.63 leaves), whose values were 7.35 and 10.60% higher than those of ‘BRS Safira’ (40.79 leaves), respectively. The difference in the number of leaves can be explained by the distinct genetic constitutions of the genotypes used in this study, with effects on growth [24]. In Experiment I, ‘BRS Rubi’ formed a larger leaf area (3117.02 cm^2^), surpassing the genotypes ‘BRS Jade’ and ‘BRS Safira’ by 17.26 and 10.01%, respectively (Figure 3C). Thus, it can be seen that ‘BRS Rubi’ had greater growth potential, which was possibly due to a higher efficiency in the photosynthetic processes, by the interception of light energy, also culminating in greater plant height and number of leaves for this cultivar [28,29].

In Experiment II, there were reductions in leaf expansion, and it was observed that water deficit in the vegetative stage of cotton, under the management strategies T_3_ (B-EV), T_6_ (C-EV), and T_9_ (BC-EV), caused reductions in the leaf area of 24.91, 24.90, and 23.28%, when compared to T_1_ (A-E0) (Figure 4). Similar results were found by Cordão et al. [25] when evaluating the effect of water deficit applied during the phenological stages of the crop on its LA.

During the second experiment (Figure 5), irrigation with a water deficit in the vegetative stage (T_3_—B-EV, T_6_—C-EV, and T_9_—BC-EV) was more severe for the genotype ‘BRS Rubi’, resulting in decreases in the number of leaves of 34.30, 21.15, and 26.63%, respectively, when compared to plants irrigated with 100% ETr. However, it can be noticed that the water deficit when applied in flowering (T_4_—B-EFL) and successively in flowering and yield formation (T_7_—C-EFR) caused less damage to the number of leaves, and the genotype ‘BRS Safira’ stood out with an NL of 53.66 and 55.00 leaves, respectively. The reduction in the number of leaves may be associated with the effects on the division and expansion of leaf cells in situations of water deficit; in addition, the genetic basis of the genotypes may be involved in conditioning the number of leaves [13,30].

According to the follow-up of the interaction between management strategies and genotypes of colored cotton for leaf dry mass (Figure 6) in Experiment I, there was no statistical difference between the strategies for ‘BRS Rubi’ and ‘BRS Jade’. However, in the T_5_ strategy (A_2_B_1_C_2_), the genotype ‘BRS Rubi’ was statistically superior to ‘BRS Jade’ and ‘BRS Safira’ by 25.58 and 30.86%, respectively. For ‘BRS Safira’, the strategies T_1_—A_1_B_1_C_1_, T_2_—A_2_B_1_C_1_, and T_3_—A_1_B_2_C_1_ were statistically superior to the others, with an LDM of 20.83, 24.45, and 21.06 g per plant (Figure 6A). In relation to LDM accumulation in Experiment I, plants that suffered from water stress in the yield formation stage had their biomass accumulation compromised, so they may be the most sensitive to water stress, as observed in the other growth variables. Reduction in biomass production due to water stress was also observed in cotton by Vidal et al. [31].

In Experiment II, for leaf dry mass, the treatments with a water deficit in Experiment I did not differ in the new cycle for the genotype ‘BRS Jade’; for ‘BRS Rubi’ and ‘BRS Safira’, the highest leaf dry mass was obtained with water deficit (40% ETr), in the T_10_ strategy (BC-EFF) for ‘BRS Rubi’ and in the T_9_ strategy (BC-EV) for ‘BRS Safira’ (Figure 6B). Higher leaf dry mass was observed in ‘BRS Rubi’, with a mean value of 27.48 g per plant, exceeding the mean values obtained in ‘BRS Jade’ and ‘BRS Safira’ by 8.98% and 6.76%, respectively (Figure 6B). Thus, a better performance of cotton genotypes in the second experiment may be due to an increase in heritability in the increments of water deficit tolerance [14,32,33].

According to the decomposition of the interaction between irrigation management strategies and genotypes of colored cotton for stem dry mass in Experiment I (Figure 7A), plants irrigated with 40% ETr in the vegetative stage, T_5_ (A_2_B_1_C_2_) strategy, had reductions of 34.06 and 24.56% in their growth for the genotypes ‘BRS Rubi’ and ‘BRS Jade’, respectively, when compared to plants under full irrigation (T_1_) throughout the cycle. However, for the genotype ‘BRS Safira’, plants that suffered water stress in the initial stages, T_2_ (A_2_B_1_C_1_), were able to recover their biomass accumulation in the flowering and yield formation stages (Figure 7A). Regarding the genotypes, ‘BRS Rubi’ was the best, with increments of 19.11 and 15.10% compared to ‘BRS Jade’ and ‘BRS Safira’, respectively, when irrigated with 40% ETr during the flowering stage (Figure 7A). According to Sá et al. [34], biomass reflects the accumulation of carbohydrates produced in photosynthesis. The reduction of photosynthetic activity caused by water stress can directly affect biomass accumulation in cotton plants. According to Gomes et al. [35], cotton has distinct characteristics, varying according to the genotype, which responds differently to local water conditions.

In Experiment II, a significant interaction was observed between irrigation management strategies and genotypes (Figure 7B). For the genotype ‘BRS Rubi’, plants that did not suffer water stress (E0) were statistically superior, with values 36.62, 39.19, 26.57, 18.31, 31.02, 29.32, 16.75, 14.08, and 12.57% higher (respectively, T_2_ to T_10_) compared to those subjected to the other management strategies. However, plants that suffered water stress in the flowering stage in the first production cycle and were subjected to full irrigation in the second production cycle (T_2_ B-E0) of the genotype ‘BRS Jade’ under the T_1_ strategy (A-E0) were 7.37, 23.29, 35.21, 27.94, 15.26, 44.16, and 18.79% superior to those under T_2_ (B-E0), T_3_ (B-EV), T_4_ (B-EFL), T_6_ (C-EV), T_7_ (C-EFR), T_9_ (BC-EV), and T_10_ (BC-EFF), respectively (Figure 7B).

According to the results of the analysis of variance (Table 2), there was only a significant effect of the interaction between the factors (strategies × genotypes) on the number of bolls and seed cotton weight in Experiment I. There were also differences between the irrigation management strategies for harvest index, water consumption, and water use efficiency in Experiment I and for the number of bolls, harvest index, water consumption, and water use efficiency in Experiment II. Regarding the cotton genotype factor, there were significant differences (*p* ≤ 0.01) in the percentage of fibers, harvest index, and water use efficiency (WUE) in Experiment I and in the number of bolls, seed cotton weight, percentage of fibers, harvest index, and water use efficiency in Experiment II (Table 2).

The number of bolls was similar among the three genotypes of colored cotton in the first experiment (Figure 8), and there were decreases of 37.51 and 60.09% caused by irrigation with a water deficit in the vegetative and flowering stages (T_6_—A_2_B_2_C_1_) in the genotypes ‘BRS Jade’ and ‘BRS Rubi’, respectively, compared to plants under full irrigation (100% ETr—T_1_). However, when the water deficit was applied only in the yield formation stage (T_4_ management—A_1_B_1_C_2_), the genotype ‘BRS Safira’ showed a higher number of bolls (16.6 bolls), which was similar to the values of plants grown in the absence of water stress (Figure 8).

Plants react to water deficit conditions through the activation of several signaling pathways and produce significant changes in gene expression and plant metabolism [36]. These reductions in the number of bolls in cotton genotypes subjected to a water stress are attributed to changes in physiological and biochemical processes at the cellular or even at the molecular level, which are associated with edaphoclimatic conditions [28,37].

When comparing the means of the genotypes ‘BRS Rubi’, ‘BRS Jade’, and ‘BRS Safira’, it was observed that, at the end of the first cycle, plants irrigated with a water deficit in the vegetative stage (A_2_B_1_C_1_) had 18.33, 22.00, and 17.33 bolls per plant, respectively, and did not differ from plants under full irrigation throughout the cycle (Figure 8). This result is an indication that their exposure to water deficit (40% ETr) during the vegetative stage causes less impact on the number of bolls per plant, that is, after the vegetative growth stage, in which sensitivity to water stress is more evident, the cotton crop becomes progressively tolerant throughout the cycle; despite a decrease in NB, the reductions are small compared to the saving of applied water, indicating tolerance of the genetic materials [28].

Based on the comparative data for the number of bolls as a function of irrigation strategies with a water deficit in Experiment II (Figure 9A), it is observed that plants grown from seeds produced with 40% ETr in the flowering stage (T_3_—A_1_B_2_C_1_) and yield formation stage (T_7_—A_1_B_2_C_2_) in Experiment I, at the end of the second production cycle, showed no reductions in NB under the strategies with a water deficit in the vegetative stage (T_3_—B-EV, T_6_—C-EV, and T_9_—BC-EV) compared to plants irrigated with 100% ETr throughout the cycle (Figure 9A). This may be related to the adaptation of the plants as they originated from seeds that had undergone water stress in the previous cycle. An explanation for that is the transmission of epigenetic characteristics due to exposure to water variations over previous cycles; when the descendant gamete and/or seed is directly subjected to factors of the same nature, genes that were silenced can be activated [15,32].

Analysis of the data related to the genotypes showed that, in the second experiment, there was a higher number of bolls in ‘BRS Rubi’ (39.9 bolls) and ‘BRS Jade’ (31.4 bolls), with increments of 32.58 and 14.33% compared to ‘BRS Safira’ (26.9 bolls), respectively (Figure 9B).

According to the means resulting from the decomposition of the MS × G interaction (Figure 10), there was a higher seed cotton weight (Experiment I) in the genotypes ‘BRS Jade’ and ‘BRS Safira’ irrigated with 40% ETr in the vegetative stage (T_2_—A_1_B_2_C_1_), with SCW of 186.93 and 166.44 g per plant, respectively. It should be noted that these strategies were also related to a higher number of bolls, indicating the link between the production components. For the genotype ‘BRS Rubi’, water deficit applied in the phenological stages of cotton caused a reduction in SCW compared to the T_1_ strategy (A_1_B_1_C_1_) (Figure 10).

Regarding seed cotton weight in Experiment II, the genotype ‘BRS Jade’ stood out with 104.36 g per plant, showing increments of 28.21 and 10.64% compared to ‘BRS Rubi’ and ‘BRS Safira’, respectively (Figure 11). According to Embrapa [38], this result is consistent with the high production potential of this genotype in a semi-arid environment, where it shows higher production when compared to ‘BRS Rubi’ and ‘BRS Safira’.

When comparing the cotton genotypes for the percentage of fibers in the two experiments (Figure 12), ‘BRS Jade’ stood out with the highest values, 35.91% in the first experiment and 37.44% in the second experiment, surpassing ‘BRS Rubi’ by 23.02 and 17.09% and ‘BRS Safira’ by 16.12 and 8.36%, respectively. Similar results were found by [39] and [40], who reported that the percentage of fiber is not affected by water stress but by genetic characteristics.

At 120 DAS, the highest harvest indices in Experiment I were obtained in plants irrigated with 100% ETr (A_1_B_1_C_1_) and when the water deficit was applied in the vegetative stage, T_2_ (A_2_B_1_C_1_) (Figure 13A). However, in the other strategies, T_3_ (A_1_B_2_C_1_), T_4_ (A_1_B_1_C_2_), T_5_ (A_2_B_1_C_2_), T_6_ (A_2_B_2_C_1_), and T_7_ (A_1_B_2_C_2_), which led to the lowest values of harvest index (3.11, 3.53, 3.43, 3.36, and 2.78%) when compared to plants irrigated with 100% ETr (A_1_B_1_C_1_), there were reductions of 21.85, 11.30, 13.81, 15.57, and 30.15%, respectively (Figure 13A); it is worth pointing out that the stress in the vegetative stage did not affect this variable, indicating tolerance to water stress. According to the interpretation of the data obtained for the harvest index (HI) in Experiment II, a similar trend is observed in plants that were grown from seeds formed under water deficit in the flowering and fruiting stages, but recovered from water stress in the new experiment with deficit irrigation (40% ETr), because there was no more effect of water deficit on the harvest index (Figure 13C). The water stress applied during the formation of seeds in the first experiment may have induced a predisposition to the expression of genes associated with improved tolerance to water deficit in the second production cycle of cotton [12,14,41].

When the genotype factor was studied separately for the harvest index, the values of ‘BRS Jade’ were 23.97 and 41.64% higher in Experiment I (Figure 13B) and 13.35 and 35.11% higher in Experiment II compared to those of the genotypes ‘BRS Safira’ and ‘BRS Rubi’, respectively (Figure 13D). Thus, it can be seen that ‘BRS Jade’ had greater production potential in both cycles. This is justified by the genetic constitution inherent to the genotype, allowing it to have superior agronomic characteristics when compared to the other genotypes studied [42], favoring the use of ‘BRS Jade’ as a genetic material suitable to sustainable agricultural exploitation [43].

On the other hand, when analyzing the development of the cotton crop, it can be observed that water consumption increases as a function of its growth until it reaches the maximum level in the reproductive development stage. According to the irrigation management strategies in Experiments I and II (Figure 14A,B), as expected, the highest water consumption was observed in plants that received 100% ETr. However, it is not possible to infer that there was a reduction in water consumption as plants under the strategies with 40% ETr were subjected to water stress. No statistical difference was observed in the different genotypes. Thus, it can be concluded that the genotypes studied have a similar water requirement.

For water use efficiency, in Experiment I, the strategies T_4_ (A_1_B_1_C_2_), T_6_ (A_2_B_2_C_1_), and T_7_ (A_1_B_2_C_2_) did not differ from each other, with increments of 22.54, 26.92, and 20.18% when compared to the strategy with 100% ETr, respectively (Figure 15A). Similarly, in Experiment II, when using the management strategy whose application of 40% ETr comprised the flowering and yield formation stages (T_10_—BC-EFF), the WUE recorded was 7.34 g mm^−1^, surpassing the WUE of plants subjected to T_8_ (BC-E0) by 32.97% at the end of the crop cycle (Figure 15C). Such an increase was more significant when compared with the other irrigation management strategies, possibly because the plants originated from seeds produced under water stress in the flowering and yield formation stages in the previous experiment; on the other hand, when water stress occurs, water use efficiency may increase due to the decrease in stomatal conductance, which affects the photosynthetic rate with greater intensity than the transpiratory rate of the leaf and, when it becomes severe, the dehydration of mesophyll cells inhibits photosynthesis, compromising mesophyll metabolism and consequently reducing water use efficiency [30].

In line with the yield data, the genotype ‘BRS Jade’ also showed higher water use efficiency, 15.70 and 6.31 g mm^−1^ in the first and second production cycles, respectively, surpassing ‘BRS Rubi’ and ‘BRS Safira’ by 37.07 and 18.02% in Experiment I (Figure 15B) and by 27.87 and 10.96% in Experiment II (Figure 15D). Water use efficiency can be increased by irrigation in the critical growth stages of cotton plants [36]. There is a reduction in yield when water deficit is applied during the flowering stage compared to the growth and production stages [26,36], demonstrating that irrigation with a controlled water deficit is an alternative for water savings in cotton plantations, especially when carried out during the most tolerant stages to water deficit [26].

## 3. Materials and Methods

### 3.1. Location of the Experiment

The research consisted of two experiments conducted from September 2020 to November 2021 under field conditions at the Center of Sciences and Agri-Food Technology (CCTA) of the Federal University of Campina Grande (UFCG), located in the municipality of Pombal, PB, Brazil, at the geographical coordinates 6°47′20′′ S latitude and 37°48′01′′ W longitude and an average altitude of 194 m. Data concerning the maximum and minimum temperature, relative humidity of air, and precipitation during the experimental period are shown in Figure 16.

### 3.2. Plant Material

The BRS Rubi colored cotton genotype was obtained by crossing a material introduced in the USA with dark brown fiber and the CNPA 7H genotype with white fiber of good quality that was widely adapted to the Northeast of Brazil. BRS Rubi has a reddish brown fiber, a crop cycle varying between 140 and 150 days, and an average productivity of 1894 kg ha^−1^ for the Brazilian semi-arid region [44].

The BRS Jade genotype has a light brown fiber, with high fiber yield potential (approximately 41%) and high productivity, exceeding 4500 kg ha^−1^. BRS Jade has a high productive potential in Cerrado and Semi-arid environments of Brazil and good fiber characteristics, with 28.6 mm fiber length, 83.7% uniformity, and a resistance of 29.2 gf/tex [45].

BRS Safira is a herbaceous or annual genotype, which can be cultivated under rainfed conditions. It results from the crossing of a dark brown fiber material introduced with CNPA 3 Precoce. Its fiber has a dark brown color, and the plants have an average height of 1.30 m and a cultivation cycle between 120 and 140 days. In the rainfed regime, it can produce, in Northeast Brazil, up to 3000 kg ha^−1^ [45].

### 3.3. Treatments and Experimental Design

In both experiments, three genotypes of naturally colored fiber cotton (‘BRS Rubi’, ‘BRS Jade’, and ‘BRS Safira’) were subjected to irrigation management strategies with a water deficit varying according to the phenological stages of the plants: (A) vegetative—the period between the emergence of the 1st true leaf and the anthesis of the 1st flower; (B) flowering—from anthesis of the 1st flower to the opening of the 1st boll; (C) yield formation—from the opening of the 1st boll to the final harvest of the bolls of the 1st cycle.

In the first experiment, the cotton genotypes were subjected to seven irrigation management strategies with two irrigation depths, 100% actual evapotranspiration (ETr) (full irrigation) and 40% ETr (water deficit), varying the phenological stages, composing the following strategies: 1—A_1_B_1_C_1_—plants under full irrigation throughout the cycle were identified by the A1 index in the phenological stages; 2—A_2_B_1_C_1_—plants under water deficit in the vegetative stage (A2 index) were irrigated with 40% ETr from 25 days after sowing—DAS to the beginning of flowering, at 61 DAS, followed by irrigation with 100% ETr until the end of the cycle; 3—A_1_B_2_C_1_—plants under water deficit in the flowering stage (B2 index), irrigated with 40% ETr from 59 DAS to the beginning of the yield formation, at 74 DAS, followed by full irrigation in the other stages; 4—A_1_B_1_C_2_—plants under water deficit in the yield formation stage (C2 index), full irrigation in the vegetative and flowering stages, and irrigation with 40% ETr from 74 DAS to the end of the cycle; 5—A_2_B_1_C_2_—plants under water deficit in the vegetative and yield formation stages (A2 and C2 indices), with full irrigation at flowering; 6—A_2_B_2_C_1_—plants under water deficit in the vegetative and flowering stages (indices A2 and B2) and full irrigation in the yield formation stage; and 7—A_1_B_2_C_2_—full irrigation only in the vegetative stage (B2 and C2 indices), and water deficit in the flowering and yield formation stages.

The experimental design was randomized blocks in a 3 × 7 factorial scheme (3 genotypes × 7 irrigation management strategies), resulting in 21 treatments, with 3 replicates and 3 plants per plot, totaling 189 plants.

In the second experiment, the irrigation management strategies resulted from the combinations of the phenological stages between the two cycles, with irrigation with water deficit (40% ETr) alternated with 100% ETr, thus characterizing cumulative water stress between the production cycles. The following letters were used for identification: A—plants grown from seeds formed with full irrigation (100% ETr) throughout the previous cycle (1-A_1_B_1_C_1_); B—plants grown from seeds formed in the first cycle with water deficit (40% ETr) in flowering (3-A_1_B_2_C_1_) and full irrigation in the other stages; C—plants grown from seeds formed when the water deficit was applied only in the yield formation stage (4-A_1_B_1_C_2_); and BC—plants grown from seeds formed in the previous cycle with water deficit in the flowering and yield formation stages (7-A_1_B_2_C_2_). The 10 irrigation management strategies used in this experiment are detailed in Table 3.

The experimental design of Experiment II was in randomized blocks, in a 3 × 10 factorial scheme, corresponding to three cotton genotypes (‘BRS Rubi’, ‘BRS Jade’, and ‘BRS Safira’) and ten irrigation strategies, with three replicates and two plants per plot, totaling one hundred eighty plants. The vegetative stage corresponded to the period between the appearance of the first true leaf and the opening of the first flower (20–61 DAS). Flowering: from anthesis of the first flower to the opening of the first boll (61–87 DAS). Yield formation: from the opening of the first boll to the final harvest of the bolls (87–117 DAS), respectively.

### 3.4. Experiment Setup and Conduction

In both experiments, the plants were grown in drainage lysimeters with a capacity of 20 L (35 cm height × 31 cm superior diameter × 20 cm inferior diameter). A 16 mm diameter transparent plastic tube was connected at the base of each lysimeter, leading to a 2.0 L container, to collect drained water and estimate water consumption by the plants. The tip of the drain inside the lysimeter was covered with a nonwoven geotextile (Bidim OP 30) to prevent clogging. The lysimeters were filled with a 0.5 kg layer of crushed stone, followed by 24.5 kg of soil material representative of the semi-arid region of the Paraíba state (properly pounded to break up clods and homogenized). The soil was collected at 0–30 cm depth (A horizon). Before starting the experiment, the soil was sampled to determine physical and chemical attributes in the Laboratory of Irrigation and Salinity (LIS) of CTRN/UFCG, according to the methodology proposed by Teixeira et al. [46], and the results are shown in Table 4.

To meet the nutritional needs, the plants were fertilized with nitrogen (N), phosphorus (P), and potassium (K), according to the fertilizer recommendation for pot experiments described by Novais et al. [47], applying 100, 300, and 150 mg kg^−^^1^ of the soil of N, P_2_O_5_, and K_2_O, respectively, in the forms of urea, monoammonium phosphate (MAP), and potassium chloride, all as top-dressing, via irrigation water, in three equal portions, at 25, 45, and 75 days after sowing (DAS) in Experiment I and at 30, 50, and 70 DAS in Experiment II. Micronutrients were applied at 20 and 25 DAS in Experiments I and II, respectively, and the applications were performed weekly, with a 1.0 g L^−^^1^ solution of Dripsol micro^®^ containing Mg—1.1%, Zn—4.2%, B—0.85%, Fe—3.4%, Mn—3.2%, Cu—0.5%, and Mo—0.05%. The lysimeters were arranged in single rows with a spacing of 1 m between plants and 0.6 m between rows.

In Experiment I, the seeds of the three cotton genotypes were provided by Embrapa Cotton, and five seeds were sown in each lysimeter at a 2 cm depth and were equidistantly distributed. After stabilization of emergence, thinning was performed to leave only one plant per pot. In Experiment II, the seeds used came from the first production cycle according to irrigation management strategies (Table 3).

Soil moisture was maintained at a level equivalent to the maximum water holding capacity (0.33 atm) in all units until the emergence of the first true leaf, when the treatments started, using water from the local supply system for irrigation (ECw = 0.3 dS m^−^^1^).

From 20 and 25 DAS in Experiments I and II, respectively, irrigation was performed daily at 5 p.m. by applying the volume of water corresponding to each treatment (40 or 100% ETr) in each container, plus a leaching fraction of 20% in plants irrigated with 100% ETr, every 7 days. The volume of water applied in each water deficit management strategy was determined based on the consumption of plants under 100% ETr, by the drainage lysimeter method [48]. For irrigation of the 40% ETr treatment, the ETr value obtained was multiplied by the percentage of evapotranspiration.

Cultural practices comprised the control of pests by preventive interventions using a hand-held compression sprayer, with a tank made of high-molar-mass polyethylene, with a volumetric capacity of 20 L. Invasive plants in the lysimeters were controlled by manual weeding during the experimental period to avoid interspecific competition for water and nutrients, favoring the full development of the crop.

Phytosanitary management was carried out preventively to control the possible appearance of pests: cotton aphid (Aphis gossypii), whitefly (Aleyrodidae), apple caterpillar (Heliothis virescens), spider mite (Tetranychus urticae), and boll weevil (Antthonomus grandis), through selective chemical products based on chlorfenapyr and cypermethrin, using 1 g for 10 L and 2.5 mL for 10 L in the preparation of the solution, respectively.

### 3.5. Variables Analyzed

At 95 DAS (Experiment I and II), the following parameters were evaluated: number of leaves (NL), plant height (PH), measured from the base of the plant to the apical meristem, stem diameter (SD), measured with a digital caliper at 5 cm from the ground, and leaf area (LA), obtained by measuring the length of the leaves, using the methodology proposed by [49], according to Equation (1)
(1)$${\mathrm{LA}}_{\mathrm{total}}=\sum_{i=1}^{n}\left(0.4322\;\times \;x^{2.3002}\right)$$
where

LA is the leaf area of each cotton leaf (cm^2^) and X is the midrib length of the respective leaf, with leaf area per plant (LA_total_) determined by summing the leaf area (LA) of all leaves.

At the end of the crop cycle, at 120 DAS in Experiment I and at 120 DAS in Experiment II, the plants were collected, separated into leaves and stems, placed in paper bags, and dried in an air circulation oven at 65 °C until reaching a constant weight; subsequently, the material was weighed on a scale with a precision of 0.001 g to obtain a dry mass of leaves (LDM) and stem (SDM). The bolls were harvested in each plant as they reached the point of harvest, quantifying the number of bolls (NB), seed cotton weight (SCW), percentage of fibers (%Fiber), and harvest index (HI), which were analyzed according to the methodology of [50].

In the same period, water consumption (WC) and water use efficiency (WUE) were quantified. Accumulated water consumption was calculated from the sum of the daily water consumption, per experimental unit, recorded during the experiments. Water consumption was calculated as the sum of the volume applied in each irrigation management strategy and cotton genotype. WUE was determined as the direct ratio between yield (seed cotton weight) and accumulated water consumption until the end of the production cycle (Equation (2)).
(2)$$\mathrm{WUE}=\frac{\mathrm{Yp}}{{WC}_{accum}}$$
where

WUE—water use efficiency in cotton cultivation (g mm^−1^);

Yp—total seed cotton weight (g plant^−1^); and

WC_accum_—accumulated water consumption per period (mm plant^−1^).

### 3.6. Statistical Analysis

The collected data were submitted to the distribution normality test (Shapiro–Wilk test). Then, the analysis of variance (*p* ≤ 0.05) was performed, where the means of irrigation strategies with water deficit were compared using the Scott–Knott cluster test (*p* ≤ 0.05), and the genotypes of cotton were compared by the Tukey test (*p* ≤ 0.05), using the statistical software SISVAR-ESAL version 5.6 [51]. Microsoft Excel 2019 tools were used to prepare the graphics.

## 4. Conclusions

Irrigation with a water deficit in the vegetative stage of cotton can be used in the first year of its cultivation. Among the genotypes, ‘BRS Jade’ is the most tolerant to the deficit regarding biomass accumulation and fiber production, regardless of the development stage. The lint cotton production of the cotton genotypes was not compromised by cumulative water stress, starting from seeds from plants subjected to water deficit in the previous cycle. In the second production cycle, colored cotton plants grown from seeds produced with water stress in the flowering and fruiting stages had increments in the number of bolls and harvest index. Future studies should elucidate the effects of epigenetic regulations of colored fiber cotton when subjected to water deficit in successive production cycles.

## Figures and Tables

**Figure 1 plants-12-02892-f001:**
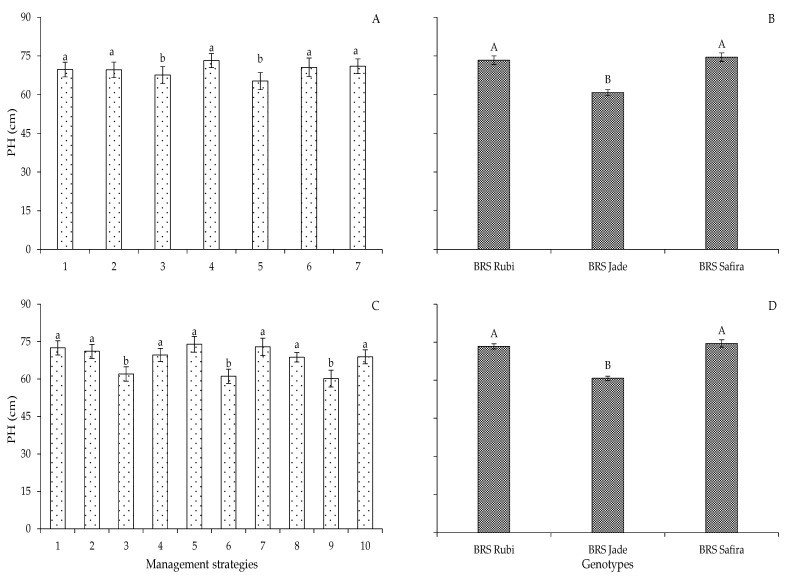
Plant height of cotton as a function of irrigation management strategies (**A**,**C**) and genotypes (**B**,**D**) in Experiments I (**A**,**B**) and II (**C**,**D**) at 96 days after sowing. In each management strategy, bars with the same lowercase letter indicate no significant difference between the means (Scott–Knott, *p* ≤ 0.05); among the genotypes, bars with the same uppercase letter do not differ by Tukey test, *p* ≤ 0.05. Experiment I: 1—no water deficit (100% ETr) during the entire cycle; water deficit (40% ETr) during 2—vegetative; 3—flowering; 4—yield formation; 5—vegetative and yield formation; 6—vegetative and flowering; and 7—flowering and yield formation stages. Experiment II: 1—no water deficit (100% ETr) with plants grown from seeds formed with full irrigation throughout the previous cycle; 2, 5, and 8—no water deficit (100% ETr) with plants grown from seeds formed with a water deficit in the vegetative, flowering, and yield formation stages, respectively; 3, 6, and 9—water deficit (40% ETr) in the vegetative stage with plants grown from seeds formed with a water deficit in the vegetative, flowering, and yield formation stages, respectively; 4, 7, and 10—water deficit (40% ETr) in the flowering, yield formation, and flowering/yield formation stages with plants grown from seeds formed with a water deficit in the vegetative, flowering, and yield formation stages, respectively.

**Figure 2 plants-12-02892-f002:**
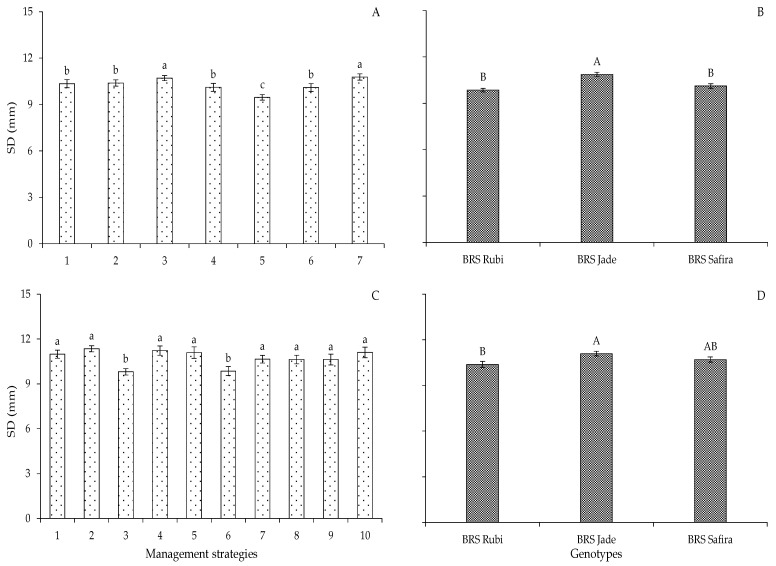
Stem diameter (SD) of cotton as a function of irrigation management strategies and genotypes in Experiment I (**A**,**B**) and Experiment II (**C**,**D**) at 95 days after sowing. In each management strategy, bars with the same lowercase letter indicate no significant difference between the means (Scott–Knott, *p* ≤ 0.05); among the genotypes, bars with the same uppercase letter did not differ from each other by Tukey test, *p* ≤ 0.05. For the explanation of the legends in the figure see the details in Figure 1.

**Figure 3 plants-12-02892-f003:**
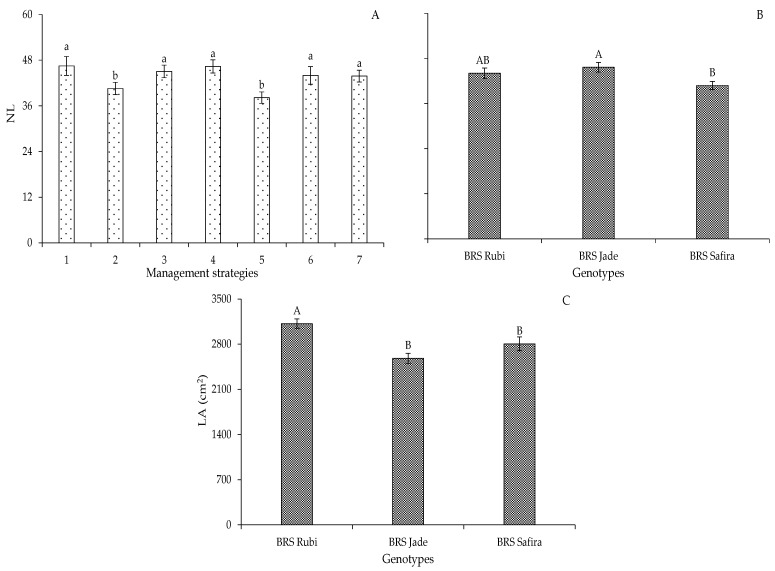
Number of leaves of cotton as a function of irrigation management strategies (**A**) and genotypes (**B**), and leaf area as a function of genotypes (**C**) at 95 days after sowing in Experiment I. In each management strategy, bars with the same lowercase letter indicate no significant difference between the means (Scott–Knott, *p* ≤ 0.05); among the genotypes, bars with the same uppercase letter did not differ from each other by Tukey test, *p* ≤ 0.05. For the explanation of the legends in the figure see the details in Figure 1.

**Figure 4 plants-12-02892-f004:**
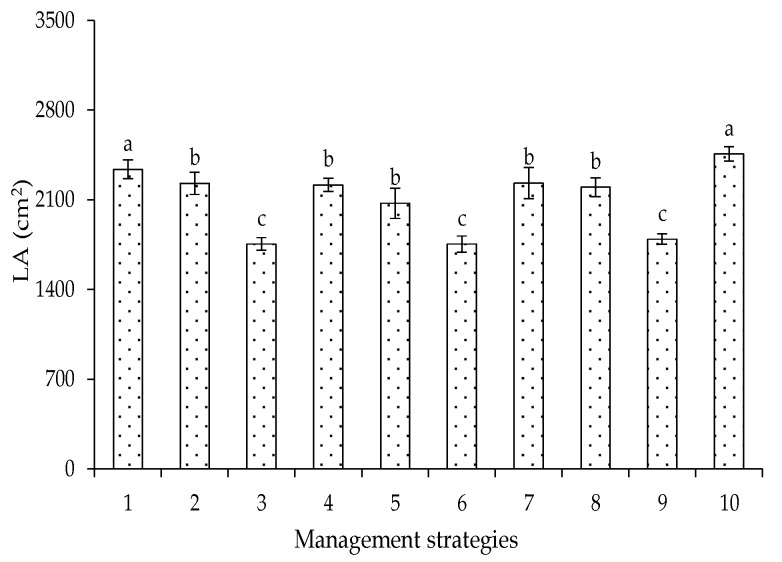
Leaf area of cotton as a function of irrigation management strategies at 95 days after sowing in Experiment II. In each management strategy, bars with the same lowercase letter indicate no significant difference between the means (Scott–Knott, *p* ≤ 0.05). For the explanation of the legends in the figure see the details in Figure 1. Leaf area formed with a water deficit in the vegetative, flowering, and yield formation stages, respectively.

**Figure 5 plants-12-02892-f005:**
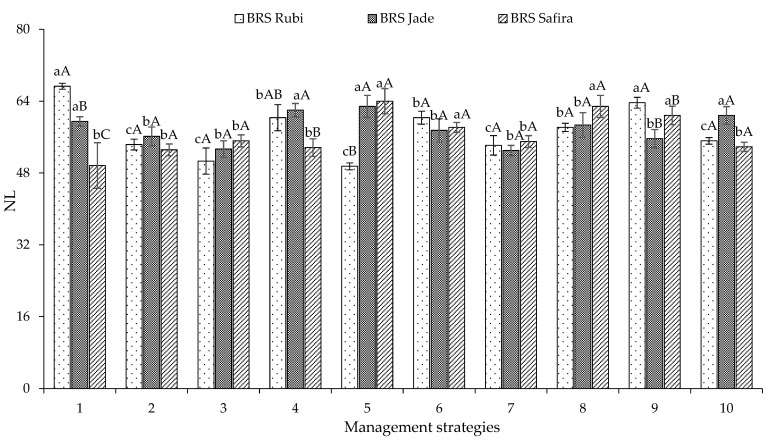
Follow-up of the interaction between genotypes and irrigation management strategies for the number of leaves (NL) of cotton at 95 days after sowing in Experiment II. In each management strategy for the same genotype, bars with the same lowercase letter indicate no significant difference between the means (Scott–Knott, *p* ≤ 0.05); among the genotypes, bars with the same uppercase letter in the same strategy do not differ from each other by Tukey test, *p* ≤ 0.05. For the explanation of the legends in the figure see the details in Figure 1.

**Figure 6 plants-12-02892-f006:**
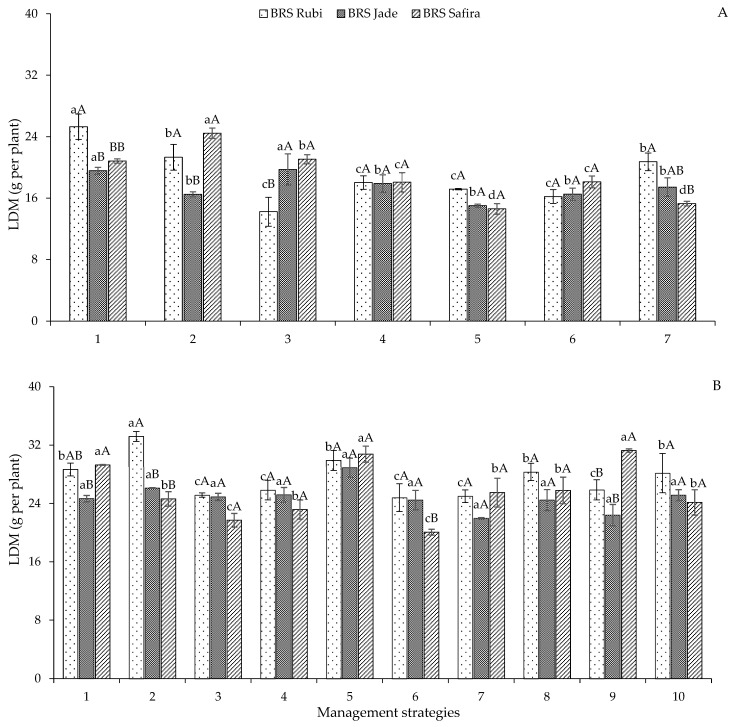
Follow-up of the interaction between irrigation management strategies and genotypes for leaf dry mass (LDM) in Experiment I (**A**) and Experiment II (**B**) at 120 days after sowing. In each management strategy for the same genotype, bars with the same lowercase letter indicate no significant difference between the means (Scott–Knott, *p* ≤ 0.05); among the genotypes, bars with the same uppercase letter in the same strategy did not differ from each other by Tukey test, *p* ≤ 0.05. For the explanation of the legends in the figure see the details in Figure 1.

**Figure 7 plants-12-02892-f007:**
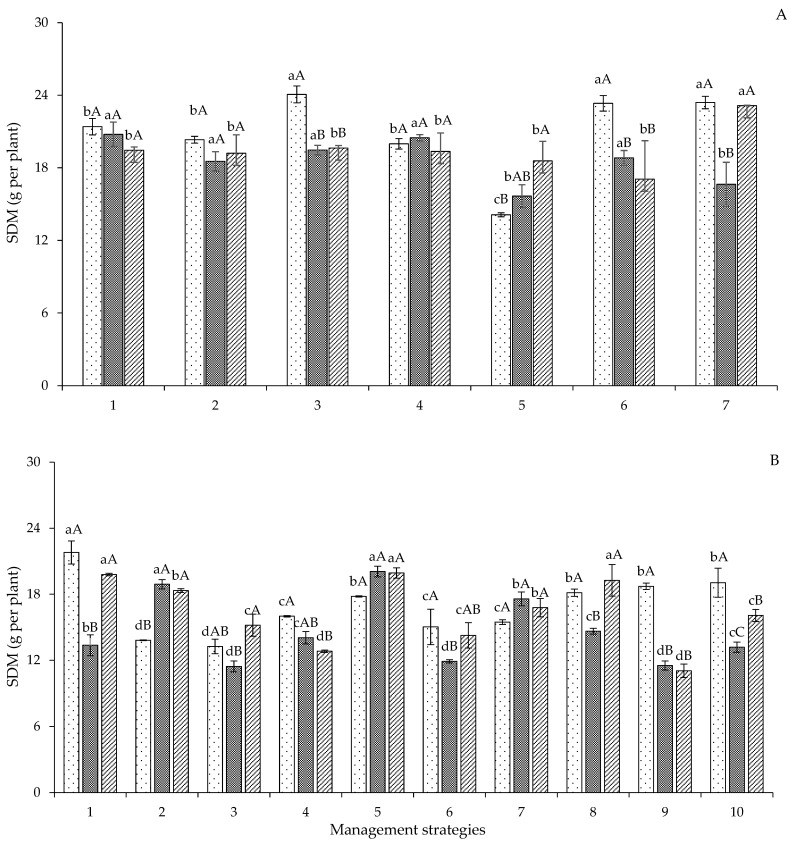
Follow-up of the interaction between irrigation management strategies and genotypes for stem dry mass (SDM) in Experiment I (**A**) and Experiment II (**B**) at 120 days after sowing. In each management strategy for the same genotype, bars with the same lowercase letter indicate no significant difference between the means (Scott–Knott, *p* ≤ 0.05); among the genotypes, bars with the same uppercase letter in the same strategy did not differ from each other by Tukey test, *p* ≤ 0.05. For the explanation of the legends in the figure see the details in Figure 1.

**Figure 8 plants-12-02892-f008:**
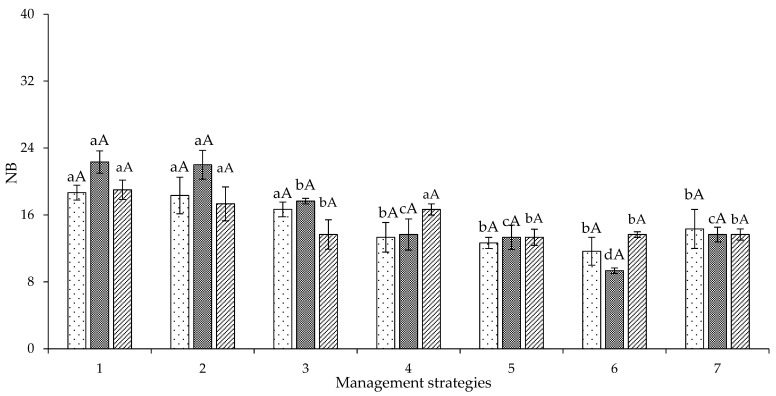
Decomposition of the interaction between genotypes and irrigation management strategies for the number of bolls (NB) of cotton at 120 days after sowing in Experiment I. In each management strategy for the same genotype, bars with the same lowercase letter indicate no significant difference between the means (Scott–Knott, *p* ≤ 0.05); among the genotypes, bars with the same uppercase letter in the same strategy did not differ from each other by Tukey test, *p* ≤ 0.05. For the explanation of the legends in the figure see the details in Figure 1.

**Figure 9 plants-12-02892-f009:**
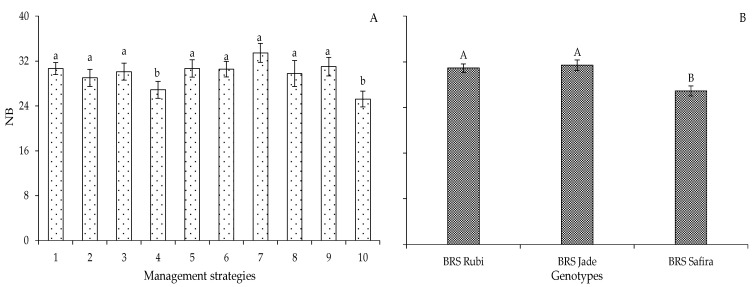
Number of bolls (NB) of cotton under different irrigation management strategies (**A**) and cotton genotypes (**B**) in Experiment II at 120 days after sowing. In each management strategy for the same genotype, bars with the same lowercase letter indicate no significant difference between the means (Scott–Knott, *p* ≤ 0.05); among the genotypes, bars with the same uppercase letter in the same strategy did not differ from each other by Tukey test, *p* ≤ 0.05. For the explanation of the legends in the figure see the details in Figure 1.

**Figure 10 plants-12-02892-f010:**
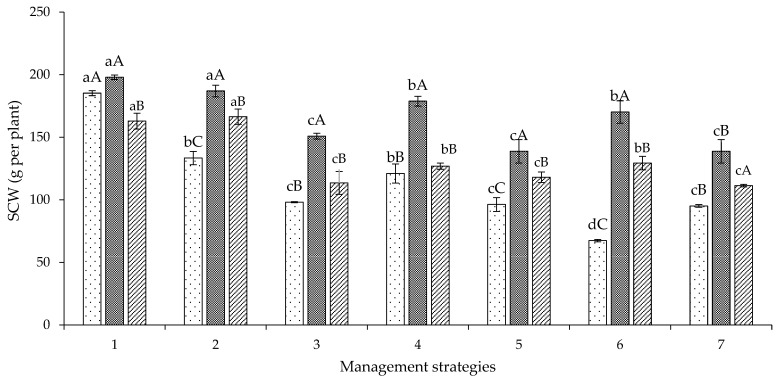
Decomposition of the interaction between genotypes and irrigation management strategies for seed cotton weight (SCW) at 120 days after sowing in Experiment I. In each management strategy for the same genotype, bars with the same lowercase letter indicate no significant difference between the means (Scott–Knott, *p* ≤ 0.05); among the genotypes, bars with the same uppercase letter in the same strategy did not differ from each other by Tukey test, *p* ≤ 0.05. For the explanation of the legends in the figure see the details in Figure 1.

**Figure 11 plants-12-02892-f011:**
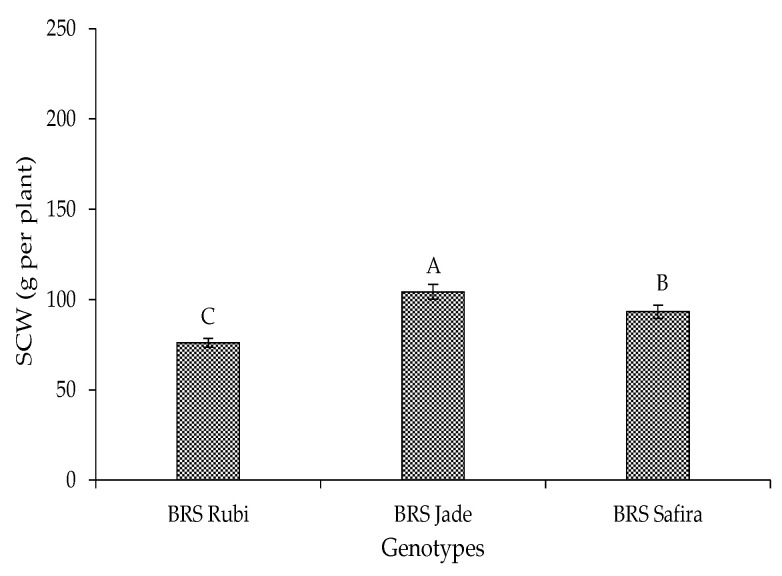
Seed cotton weight (SCW) of cotton genotypes at 120 DAS in Experiment II. Bars with the same uppercase letter do not differ from each other by Tukey test, *p* ≤ 0.05.

**Figure 12 plants-12-02892-f012:**
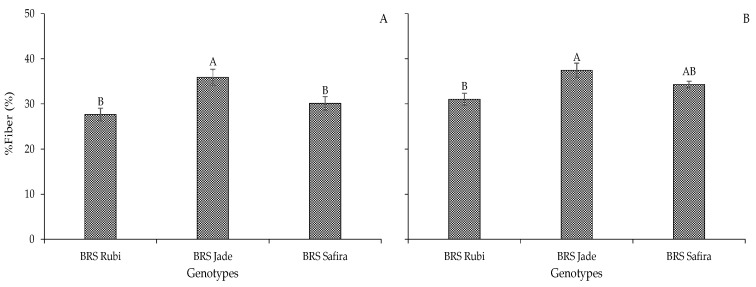
Mean percentage of fibers (%Fiber) of cotton genotypes in Experiment I (**A**) and Experiment II (**B**) subjected to different irrigation strategies. Bars with the same uppercase letter do not differ from each other by Tukey test, *p* ≤ 0.05.

**Figure 13 plants-12-02892-f013:**
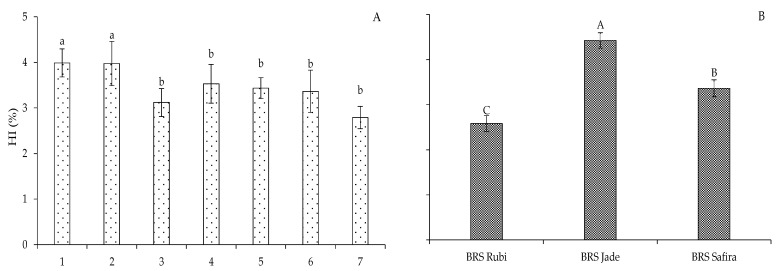
Harvest index (HI) of cotton under different irrigation management strategies (**A**,**C**) and genotypes (**B**,**D**) in Experiments I (**A**,**B**) and II (**C**,**D**). In each management strategy, bars with the same lowercase letter indicate no significant difference between the means (Scott–Knott, *p* ≤ 0.05); among the genotypes, bars with the same uppercase letter did not differ from each other by Tukey test, *p* ≤ 0.05. For the explanation of the legends in the figure see the details in Figure 1.

**Figure 14 plants-12-02892-f014:**
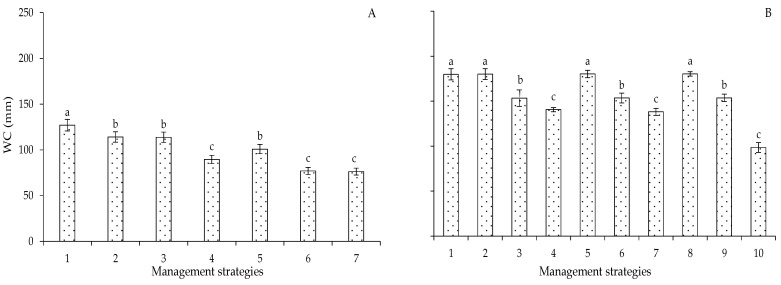
Mean water consumption (WC) of cotton under different irrigation management strategies in Experiment I (**A**) and Experiment II (**B**) at 120 days after sowing. In each management strategy, bars with the same lowercase letter indicate no significant difference between the means (Scott–Knott, *p* ≤ 0.05). For the explanation of the legends in the figure see the details in Figure 1.

**Figure 15 plants-12-02892-f015:**
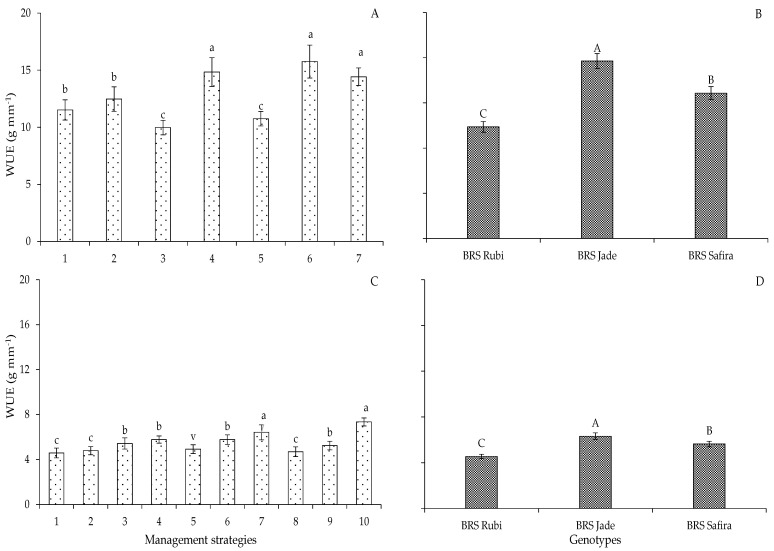
Water use efficiency (WUE) of cotton under different irrigation management strategies (**A**,**C**) and genotypes (**B**,**D**) in Experiments I (**A**,**B**) and II (**C**,**D**). In each management strategy, bars with the same lowercase letter indicate no significant difference between the means (Scott–Knott, *p* ≤ 0.05); among the genotypes, bars with the same uppercase letter did not differ from each other by Tukey test, *p* ≤ 0.05. For the explanation the legends in the figure see the details in Figure 1.

**Figure 16 plants-12-02892-f016:**
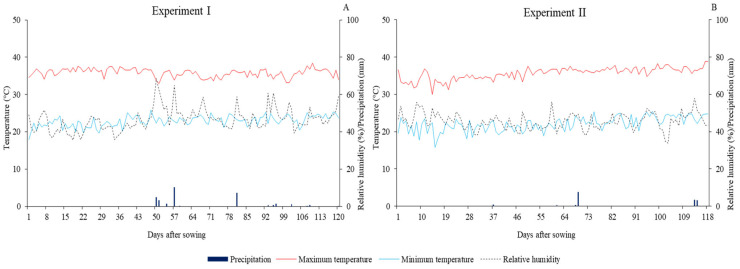
Data concerning the maximum and minimum temperature, relative humidity of air, and precipitation during Experiment I (**A**) and Experiment II (**B**).

**Table 1 plants-12-02892-t001:** Summary of the analysis of variance for plant height (PH), stem diameter (SD), number of leaves (NL), leaf area (LA), dry mass of leaves (LDM), and stem (SDM) as a function of different irrigation management strategies and cotton genotypes in Experiments I and II.

Variables	DF	Mean Squares					
		Experiment I					
		PH	SD	NL	LA	LDM	SDM
Strategies (S)	6	57.88 **	1.77 **	85.70 **	306,614.41 ^ns^	42.37 **	26.06 **
Genotypes (G)	2	1216.38 **	5.77 **	127.70 *	1,533,690.29 **	14.41 *	28.91 **
S × G	12	13.52 ^ns^	0.25 ^ns^	10.90 ^ns^	131,676.48 ^ns^	21.95 **	14.78 **
Blocks	2	801.49 **	0.26 ^ns^	2.35 ^ns^	172,268.89 ^ns^	4.62 ^ns^	1.52 ^ns^
Residual	40	17.59	0.23	34.45	148,975.87	3.28	4.08
CV (%)		6.03	4.74	13.50	4.74	9.80	10.27
		Experiment II					
Strategies (S)	9	69.83 *	2.63 **	70.87 **	573,779.38 **	35.73 **	45.22 **
Genotypes (G)	2	1695.18 **	3.80 **	18.85 ^ns^	62,178.84 ^ns^	49.57 **	30.03 **
S × G	18	28.47 ^ns^	0.71 ^ns^	66.54 **	43,513.00 ^ns^	19.97 **	16.56 **
Blocks	2	62.25 ^ns^	1.08 ^ns^	1.30 ^ns^	503,244.40 **	1.08 ^ns^	1.71 ^ns^
Residual	58	35.80	0.73	12.88	41,845.31	4.59	1.48
CV (%)		8.57	8.01	6.27	15.00	8.24	7.57

^ns^, *, **: not significant and significant at *p* ≤ 0.05 and *p* ≤ 0.01, respectively, by the F test.

**Table 2 plants-12-02892-t002:** Summary of the analysis of variance for the number of bolls (NB), seed cotton weight (SCW), percentage of fibers (%Fiber), harvest index (HI), water consumption (WC), and water use efficiency (WUE) as a function of different irrigation management strategies and cotton genotypes in Experiments I and II at 120 days after sowing.

Variables	DF	Mean Squares					
		Experiment I					
		NB	SCW	%Fiber	HI	WC	WUE
Strategies (S)	6	93.32 **	60,333.53 **	13.04 ^ns^	1.68 *	34.77 **	44.43 **
Genotypes (G)	2	6.33 ^ns^	14,702.10 **	378.35 **	17.96 **	2.8 ^ns^	177.99 **
S × G	12	11.70 **	754.43 **	14.48 ^ns^	0.58 ^ns^	9.47 ^ns^	3.59 ^ns^
Blocks	2	28.42 *	311.08 *	11.49 ^ns^	0.25 ^ns^	2.84 ^ns^	3.50 ^ns^
Residual	40	5.74	80.40	15.64	0.55	5.68	2.45
CV (%)		15.44	6.52	12.67	21.63	5.1	12.22
		Experiment II					
Strategies (S)	9	46.91 **	264.38 ^ns^	43.92 ^ns^	0.46 *	6201.3 **	6.74 **
Genotypes (G)	2	182.50 **	6628.82 **	307.34 **	6.47 **	0.0004 ^ns^	23.67 **
S × G	18	14.61 ^ns^	256.38 ^ns^	48.08 ^ns^	0.28 ^ns^	0.0004 ^ns^	0.85 ^ns^
Blocks	2	158.70 **	2484.06 ^ns^	72.84 ^ns^	1.07 **	0.0004 ^ns^	7.76 **
Residual	58	15.13	283.03	46.63	0.20	0.0004	1.00
CV (%)		13.08	18.52	19.93	20.38	5.00	18.19

^ns^, *, **: not significant and significant at *p* ≤ 0.05 and *p* ≤ 0.01, respectively, by the F test.

**Table 3 plants-12-02892-t003:** Irrigation management strategies in the phenological stages of cotton in the second experiment, with information on the treatments from which the seeds were collected in the first cycle.

Irrigation Management Strategies		Phenological Stages		
		Vegetative (A)	Flowering (B)	Yield Formation (C)
Experiment I ^1^	Experiment II ^2^	ETr (%)		
A_1_B_1_C_1_	A-E0	100	100	100
A_2_B_1_C_1_	B-E0	100	100	100
	B-EV	40	100	100
	B-EFL	100	40	100
A_1_B_2_C_1_	C-E0	100	100	100
	C-EV	40	100	100
	C-EFR	100	100	40
A_1_B_2_C_2_	BC-E0	100	100	100
	BC-EV	40	100	100
	BC-EFF	100	40	40

Experiment I: ^1^ A_1_, B_1_, and C_1_: no water deficit in the vegetative, flowering, and fruiting stages, and denominations A_2_, B_2_, and C_2_ refer to water deficit in the vegetative, flowering, and fruiting stages, respectively. Experiment II: ^2^ A—seeds from plants irrigated with 100% ETr in the first experiment; B, C, and BC—seeds from plants irrigated with 40% ETr in the vegetative, flowering, fruiting, and flowering/fruiting stages, respectively; E0—100% ETr throughout the cycle; EV, EFL, EFR, and EFF: 40% ETr in the vegetative, flowering, fruiting, and flowering/fruiting stages, respectively.

**Table 4 plants-12-02892-t004:** Chemical and physical characteristics of the soil used in the experiments before the application of the treatments.

Experiment I										
Bulk Density	Total Porosity	Moisture (%)		Available Water	Sorption Complex					
					Ca^2+^	Mg^2+^	Na^+^	K^+^	pH_sp_	EC_se_
kg dm^−3^	%	0.33 atm	15 atm	%	cmol_c_ kg^−1^				-	dS m^−1^
1.37	48.88	15.01	5.81	9.20	6.4	4.11	0.1	0.8	7.76	0.22
Experiment II										
1.33	49.81	16.75	6.50	10.25	6.57	5.28	0.22	5.41	7.72	0.45

Ca^2+^ and Mg^2+^ extracted with 1 M KCl at pH 7.0; Na^+^ and K^+^ extracted with 1 M NH_4_OAc; P—Mehlich 1 extractant; pH_sp_—pH of the saturation paste; and EC_se_—electrical conductivity of the saturation extract. Soil moisture at 0.33 and 15 atm corresponds to field capacity and permanent wilting point, respectively.

## Data Availability

Not applicable.

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
