# Peer review of "Irrigation Strategies with Controlled Water Deficit in Two Production Cycles of Cotton"

_plants, 2023, doi:10.3390/plants12162892_

Round 1

Reviewer 1 Report

This is an interesting article. Overall, the language is relatively standardized, the level of scientific research is relatively high, and the experimental workload is sufficient. I suggest accepting the manuscript after modification, but before accepting it, the following issues need to be addressed:

1.I suggest supplementing the existing research progress in the introduction section, which will help improve the completeness of your manuscript. At present, the entire introduction section is relatively short, and I suggest that you reorganize the content of this section.

2. There are a total of 16 figures in this manuscript, which is a bit too many. I suggest merging some of them or submitting them as appendix.

3. The types of the results and discussion sections are too single, all of which are bar charts. Please enrich the types of the following figures, as this will increase the richness of your manuscript.

4. Please provide some data based description of the results in the abstract and conclusion section. Currently, there are no quantitative results available.

5.Please provide information on the data analysis software you are using in section 3.6.

6.What are the reasons for the selection of three different genotypes of cotton in this manuscript? Are there any fundamental differences between them? I suggest adding some relevant descriptions.

7.What is the next plan for this experiment? Will the scope of the experiment continue to be expanded? Will it shift from different water usage to different fertilization rates? I suggest you add some future outlook sections, which will be meaningful for improving your manuscript.

Some modifications need to be made.

Author Response

Caraúbas, RN

Jul, 29, 2023

Reference: Plants - 2508726 - Response to Review Report 1

Dear Editor

The authors are grateful to you and the unanimous Reviewers for the positive and constructive comments and suggestions on our manuscript entitled “Irrigation Strategies with Controlled Water Deficit in Two Production Cycles of Cotton”. The authors would like to inform you that a thorough revision of the manuscript was made, incorporating the suggestions and adopting the text according to the comments. Attached is the revised version of the manuscript. All changes in the text are highlighted in red color.

The authors remain at your disposal for any further information and explanation.

The responses/clarifications to the issues raised by the Reviewer 1/Editor are presented below:

REVIEWER 1

This is an interesting article. Overall, the language is relatively standardized, the level of scientific research is relatively high, and the experimental workload is sufficient. I suggest accepting the manuscript after modification, but before accepting it, the following issues need to be addressed:

  1. I suggest supplementing the existing research progress in the introduction section, which will help improve the completeness of your manuscript. At present, the entire introduction section is relatively short, and I suggest that you reorganize the content of this section.

Response: The introduction was reformulated in the revised version of the manuscript, incorporating and adapting the reviewers' suggestions.

  1. There are a total of 16 figures in this manuscript, which is a bit too many. I suggest merging some of them or submitting them as appendix.

Response:  Interaction figures were reformulated in the revised version of the manuscript, reducing the number of figures.

  1. The types of the results and discussion sections are too single, all of which are bar charts. Please enrich the types of the following figures, as this will increase the richness of your manuscript.

Response: Authors would like to inform, that the factors analyzed in this study are qualitative factors, for this reason, it was not possible to use other types of graphs, such as regression.

  1. Please provide some data based description of the results in the abstract and conclusion section. Currently, there are no quantitative results available.

Response:  The abstract was reformulated in the revised version of the manuscript, incorporating the reviewers' suggestions. Considering the norms of the Journal for the abstract (maximum 200 words), it was not possible to present all the results obtained in the research. It is noteworthy that the factors analyzed in this study are qualitative, making it impossible to present the results in quantitative terms.

  1. Please provide information on the data analysis software you are using in section 3.6.

Response: Section 3.6 was redrafted in the revised version of the manuscript, taking into account the reviewers' suggestions.

  1. What are the reasons for the selection of three different genotypes of cotton in this manuscript? Are there any fundamental differences between them? I suggest adding some relevant descriptions.

Response:  A detailed description of each genotype was included in the revised version of the manuscript (Section 3.2. Plant Material), highlighting the characteristics that were taken into consideration in the study.

  1. What is the next plan for this experiment? Will the scope of the experiment continue to be expanded? Will it shift from different water usage to different fertilization rates? I suggest you add some future outlook sections, which will be meaningful for improving your manuscript.

Response:  As described between lines 686 and 688 of the revised version of the manuscript, future studies should elucidate the effects of epigenetic regulation of colored fiber cotton when subjected to water deficit in successive production cycles.

Yours sincerely,

Reginaldo Gomes Nobre

Reviewer 2 Report

The reviewed manuscript entitled “Irrigation Strategies with Controlled Water Deficit in Two Pro- 2 duction Cycles of Cotton” The authors were interested in studying evaluate the morphology, production and water relations of genotypes of colored-fiber cotton plants subjected to irrigation strategies with water deficit 20 in the phenological stages, as well as the influence of water deficit on seeds of these genotypes in a 21 new production cycle, using irrigation management strategies. I think this manuscript can only be published after addressing several major issues as following: 

There are many grammatical and grammatical errors. I suggest the authors re-check the entire manuscript linguistically and get revisions by native English speakers.

1- Abstract section: In the abstract, the research gap should be improved to strengthen the motivation of the work.

2- Abstract lacks clear results from the potting experiment and biological control of the disease

3- The novelty of the study needs to be highlighted compared to other similar studies.

4- Introduction part:  Must contains the whole background regarding the targeted problem and how to solve that problem with comparison with literature review; please check and revised accordingly.

5- Materials and methods section:   The material part lacks references   -Must contains recent and related references with more details to be beneficial to broad scientific readers.

6- All abbreviations used should be mentioned in the place of their first mention followed by an abbreviation and then only the abbreviation is written, revised all manuscript. 

7- Are morphological characteristics measured? please explain?

8- Indications of the occurrence of resistance against water deficiency were measured?

9- The methods are not clear and lack recent references. Please review them carefully

10- Statistical Analysis not clear and lack recent references.

11- Discussion is written very poorly and very long. The discussion should be a bit in-depth, therefore, discussion and the writing should be improved to make the manuscript easy to follow, which made this work so strong and attractive.

12- There is no conclusion; I think author should try to link better their work; I mean, the results should be quantitatively reported to present these potential applications better.

Author Response

Caraúbas, RN

Jul, 29, 2023

Reference: Plants - 2508726 - Response to Review Report 2

Dear Editor

The authors are grateful to you and the unanimous Reviewers for the positive and constructive comments and suggestions on our manuscript entitled “Irrigation Strategies with Controlled Water Deficit in Two Production Cycles of Cotton”. The authors would like to inform you that a thorough revision of the manuscript was made, incorporating the suggestions and adopting the text according to the comments. Attached is the revised version of the manuscript. All changes in the text are highlighted in red color.

The authors remain at your disposal for any further information and explanation.

The responses/clarifications to the issues raised by the Reviewer 2/Editor are presented below:

REVIEWER 2

The reviewed manuscript entitled “Irrigation Strategies with Controlled Water Deficit in Two Pro- 2 duction Cycles of Cotton” The authors were interested in studying evaluate the morphology, production and water relations of genotypes of colored-fiber cotton plants subjected to irrigation strategies with water deficit in the phenological stages, as well as the influence of water deficit on seeds of these genotypes in a new production cycle, using irrigation management strategies. I think this manuscript can only be published after addressing several major issues as following:

There are many grammatical and grammatical errors. I suggest the authors re-check the entire manuscript linguistically and get revisions by native English speakers.

  1. Abstract section: In the abstract, the research gap should be improved to strengthen the motivation of the work.

Response: The manuscript was thoroughly revised. The abstract was reformulated, in the revised version of the manuscript, incorporating the reviewer's suggestion, as can be seen between lines 19 and 21.

  1. Abstract lacks clear results from the potting experiment and biological control of the disease.

Response:  Authors would like to inform the Reviewer that taking into consideration the journal's norms for the abstract, which establishes a limit of  200 words, it was not possible to add this information. However, the information was included in the methodology of the revised version of the manuscript, as can be seen between lines 632 and 636.

  1. The novelty of the study needs to be highlighted compared to other similar studies

Response:  The originality of the study was highlighted in the revised version of the manuscript, as can be seen between lines 68 and 81.

  1. Introduction part: Must contains the whole background regarding the targeted problem and how to solve that problem with comparison with literature review; please check and revised accordingly.

Response:  The introduction was reformulated in the revised version of the manuscript, incorporating and adapting the reviewers' suggestions.

  1. Materials and methods section: The material part lacks references -Must contains recent and related references with more details to be beneficial to broad scientific readers.

Response: It was not possible to update the references of the material and methods, as the methodologies are standard and old, making it impossible to update them. However, the recente publications were introduced in the Introduction and Result and Discussion, with the insertion of 8 references from 2023.

  1. Are morphological characteristics measured? please explain?

Response: As described between lines 638 and 657 of the revised version of the manuscript, the morphological characteristics were evaluated through plant height, stem diameter, number of leaves, leaf area, dry mass of leaves, and stem.

  1. Indications of the occurrence of resistance against water deficiency were measured?

Response: In this study, no indice of resistance to water deficiency was measured.

  1. The methods are not clear and lack recent references. Please review them carefully.

Response:   It was not possible to update the references of the material and methods, as the methodologies are standard and old, making it impossible to update them. However, the update was carried out in the Introduction and Result and Discussion section, with the insertion of 8 references from 2023.

  1. Statistical Analysis not clear and lack recent References.

Response:   Section 3.6 was redrafted in the revised version of the manuscript, taking into account the reviewers' suggestions.

  1. Discussion is written very poorly and very long. The discussion should be a bit in-depth, therefore, discussion and the writing should be improved to make the manuscript easy to follow, which made this work so strong and attractive.

Response: Authors would like to inform that, in the revised version of the manuscript, the interaction figures were reformulated in order to reduce the number of figures and facilitate the reader's understanding. However, it must be considered that this research was carried out through two field experiments, which makes the presentation of results and the discussion lengthy.

  1. There is no conclusion; I think author should try to link better their work; I mean, the results should be quantitatively reported to present these potential applications better.

Response:   Authors would like to inform Reviewer that the factors analyzed in this study are qualitative factors (water deficit irrigation management strategies and colored cotton genotypes), for this reason, it was not possible to report the results quantitatively.

Yours sincerely,

Reginaldo Gomes Nobre

Round 2

Reviewer 2 Report

 Accept in present form